# An Overall Deformation Monitoring Method of Structure Based on Tracking Deformation Contour

**Xi Chu [1], Zhixiang Zhou [1,2,\*], Guojun Deng [1], Xin Duan [1] and Xin Jiang [1]**

[1] State Key Laboratory of Mountain Bridge and Tunnel Engineering, Chongqing Jiaotong University, Chongqing 400074, China; chuxi1986@163.com (X.C.); dengguojun_cqjtu@163.com (G.D.); duanxin_cqjtu@163.com (X.D.); jiangxin_cqjtu@163.com (X.J.)

[2] College of Civil and Transportation Engineering, Shenzhen University, Shenzhen 518060, China

\* Correspondence: zhixiangzhou@cqjtu.edu.cn



**Featured Application: This study aims at the lack of sufficient data supporting structural damage identification, which is a general issue in traditional single-point measurement method. A novel method has been proposed for structural deformation monitoring based on digitalized photogrammetry, with improved efficiency and reduced cost. The method can be applied in the overall deformation monitoring of engineering structures, such as bridges. Furthermore, the method can provide a solid foundation for the estimation of structural health state.**

**Abstract:** In structural deformation monitoring, traditional methods are mainly based on the deformation data measured at several individual points. As a result, only the discrete deformation, not the overall one, can be obtained, which hinders the researcher from a better and all-round understanding on the structural behavior. At the same time, the surrounding area around the measuring structure is usually complicated, which notably escalates the difficulty in accessing the deformation data. In dealing with the said issues, a digital image-based method is proposed for the overall structural deformation monitoring, utilizing the image perspective transformation and edge detection. Due to the limitation on camera sites, the lens is usually not orthogonal to the measuring structure. As a result, the obtained image cannot be used to extract the deformation data directly. Thus, the perspective transformation algorithm is used to obtain the orthogonal projection image of the test beam under the condition of inclined photography, which enables the direct extraction of deformation data from the original image. Meanwhile, edge detection operators are used to detect the edge of structure's orthogonal projection image, to further characterize the key feature of structural deformation. Using the operator, the complete deformation data of structural edge are obtained by locating and calibrating the edge pixels. Based on the above, a series of load tests has been carried out using a steel–concrete composite beam to validate the proposed method, with the implementation of traditional dial deformation gauges. It has been found that the extracted edge lines have an obvious sawtooth effect due to the illumination environment. The sawtooth effect makes the extracted edge lines slightly fluctuate around the actual contour of the structure. On this end, the fitting method is applied to minimize the fluctuation and obtain the linear approximation of the actual deflection curve. The deformation data obtained by the proposed method have been compared with the one measured by the dial meters, indicating that the measurement error of the proposed method is less than 5%. However, since the overall deformation data are continuously measured by the proposed method, it can better reflect the overall deformation of the structure, and moreover the structural health state, when compared with the traditional "point" measurements.

**Keywords:** structural engineering; overall deformation monitoring; perspective transformation; edge detection; close-range photogrammetry

## 1. Introduction

During the service life, engineering structures are subjected to various inherent deterioration processes of structure such as corrosion, fatigue, material creep, and so on. As a result, the deformation of the degraded structure will deviate from the original one. On this end, the structural deformation in eventually used as an important index in the structural health monitoring [1,2]. For instance, the external load and deformation of the structure system generally follows the below relation:

$$\{d\} = [K]^{-1}\{f\}$$

where $\{d\}$ stands for the deformation state; $[K]$ is the stiffness matrix of the structure; $\{f\}$ represent the effect induced by the external load. When any damage or deterioration occurs in the structure, the stiffness matrix $[K]$ will change correspondingly, which in turns lead to the inevitable change in the deformation state $\{d\}$. Therefore, the change in the deformation state can be utilized to evaluate the health state of the structure. The present study focuses on the direct extraction of the overall deformation, rather than approximating the deformation through the data measured at several discrete points. The major advantage of the overall deformation data is to eliminate the error in structural health evaluation caused by insufficient measurement.

Traditionally, the structural deformation can be measured by leveling, total station, GPS, vibration sensors, and other equipment. At present, these methods can accurately and rapidly measure the deformation information of structures. However, only the limited key points of the targeting structure can be measured using the above methods, which often lead to insufficient data and, moreover, the insensitivity to structural deterioration [3]. Obviously, the direct solution is to largely increase the number of sensors installed on the structure. However, it is both time and budget consuming, which is not applicable in engineering practices. Alternatively, the digital image full-field structural morphology measurement can be a very ideal solution, which can take the advantages of both the structural damage identification method and digital image processing technology. Therefore, it is very crucial to effectively utilize the structural image features to extract the full-field deformation information of the structure.

In recent years, digital image processing technology is eventually employed to measure the overall deformation of the structure. As a kind of remote sensing technique, photogrammetry does not need any contact with the objects, and this can be a great advantage in the deformation monitoring of structures. "Photogrammetry" is to set up a base station in a stable area on the front of the target, and then shoot the target, so as to get the shape and motion state of the target according to the image [4]. According to different imaging distances, photogrammetry can be divided into "space photogrammetry", "close-range photogrammetry", and "microscopic photogrammetry" [5]. In structural deformation monitoring, close-range photogrammetry has broad prospects for development [6], and "Close-range photogrammetry" means that the distance between the base station and the measured structure is within 300 m [7]. Feng et al. [8] presents a comprehensive review on the recent development of computer vision-based sensors for structural displacement response measurement and their applications for SHM. Importation issues critical to successful measurement are discussed in detail, including how to convert pixel displacements to physical displacements, how to achieve sub-pixel resolutions, and what to cause measurement errors and how to mitigate the errors. However, the article also clearly points out that in many respects, the vision-based sensor technology is still in its infancy. The majority studies have still been focused on measurements of small-scale laboratory structures or field measurements of large structures at a limited number of points for a short period of time. Rolands Kromanis et al. [9] introduces a low-cost robotic camera system (RCS) for accurate measurement collection of structural response. The low-cost RCS provides very accurate vertical displacements. The measurement error of the RCS is 1.4%. Serena Artese et al. [10] proposed a bridge monitoring system, which combines camera and laser indicator; the elastic line inclination is measured by analyzing the single frames of an HD video of

the laser beam imprint projected on a flat target. The inclination of the elastic line at the support was obtained with a precision of 0.01 mrad. Ghorban et al. [11] measured the overall deformation of the masonry wall subjected to cyclic loads, using the 3D image correlation technology. The displacement, rotation, and interface slip between the reinforced concrete column and masonry were measured. Wang et al. [12] used the close-range photogrammetry technology to monitor the displacement of tunnel caverns. The measured results were compared with the values measured by mechanical convergence meter, and the difference between the two methods is no more than ±2 mm at the measuring distance of 8 m. This accuracy meets the requirement of general tunnel deformation monitoring. Reference [13] studied the application of sub-pixel displacement measurement method in soil strain monitoring. Based on the spatial correlation function iteration, the sub-pixel displacement of soils was measured. Zang et al. [14] applied the close-range photogrammetry technology in measuring bridge deflection and proved that a desirable accuracy can be achieved, i.e., ±1 mm. However, the accuracy is greatly affected by the positioning of the artificial marking points required by the method [15]. Although the above studies validated the feasibility of the application of close-range photogrammetry technology in structural deformation monitoring, the above methods are still unable to measure the structural overall deformation.

In order to explore the feasibility of photogrammetry in structural overall deformation monitoring, Ivan Detchev et al. [16] explored the use of consumer-grade cameras and projectors for the deformation monitoring of structural elements. A low-cost digital camera deformation monitoring system is proposed. Static load tests of concrete beams are carried out in the laboratory. The experiments proved that it was possible to detect sub-millimeter-level overall deformations given the used equipment and the geometry of the setup. However, this technology requires high texture characteristics of the structure surface and needs to project random pattern on the structure surface, which is difficult to achieve in the actual bridge structure deformation monitoring. On another hand, the close-range photogrammetry requires the measuring equipment to be located in the orthogonal projection position of the measured surface, which is usually difficult in engineering practice. Taking the bridge structure as an example, the surroundings near the bridge are usually complex, such as mountains, rivers, and trees, which makes it difficult for the camera to maintain the orthogonal projection position with respect to the measuring bridge.

In the deformation monitoring of structures, environmental factors must be considered. The complex geographical conditions of the structures means the photogrammetric camera is unable to work in the ideal measuring position. Therefore, it is necessary to study a new photogrammetric method for the overall deformation of structures under the condition that the camera is in the inclined position. In view of the actual needs of the structure deformation monitoring, this paper studies the overall deformation monitoring method under the condition of tilt photography. The steel truss concrete composite beam specimens were made in the laboratory. The static deformation images of the specimens were obtained by oblique photography. The overall deformation of the specimens was obtained by perspective transformation and edge detection technology. The error sources of this method were analyzed. This research is a comprehensive application of photogrammetry and digital image processing technology in the field of structure deformation monitoring. Its research foundation has been carefully verified and published in many publications [17,18]. The research results can alleviate the problem of insufficient deformation data in damage identification. In addition, compared with traditional photogrammetric methods, this study also highlights the advantages of flexible placement of camera positions in actual measurement work.

## 2. Orthogonal Projection and Global Deformation Acquisition Method of Structures

### 2.1. Perspective Transformation of Digital Image

On the basis of unchanged image content, the image pixel position is transformed, which is called image geometric transformation [19]. It mainly includes translation,

rotation, zooming, reflection, and slicing. Usually, compound transformations, such as the perspective transformation, can be divided into a series of basic transformations. According to the perspective principle [20], when photographed under the condition of non-orthogonal projection, the image of the measured structure will deform. As a result, the true shape of the structure can be obtained only when the camera is in the orthogonal projection position of the measured surface. The process of mathematical transformation from oblique projection center to orthogonal projection center is called the perspective transformation. Figure 1 illustrates the basic model of perspective transformation.

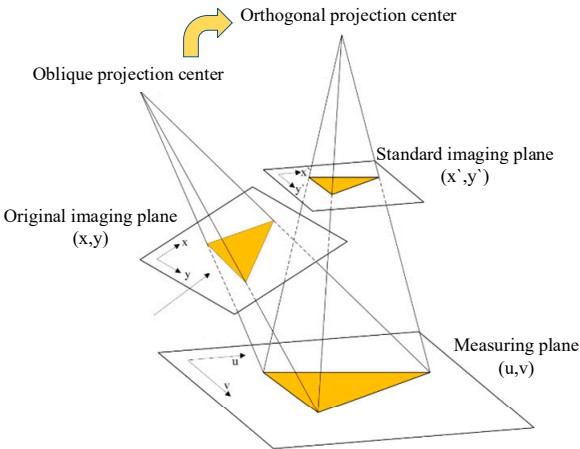

**Figure 1.** Perspective transformation.

A 3D Cartesian coordinate system can be established, in which the projection center of the camera is selected as the origin, called the camera coordinate. Meanwhile, the image plane is set as the x-y plane, and the focus of the plane is located at $[0, 0, f]$ $(f > 0)$. A 2D Cartesian coordinate system can be established on the plane where the object is measured, called the measuring coordinate. The origin of the measuring coordinate system is $[x_0, y_0, z_0]^T$ in the camera coordinate. The unit vectors in the $x$-axis direction are $[u_1, u_2, u_3]^T$, the unit vectors in the $y$-axis direction are $[v_1, v_2, v_3]^T$, and the vector relation can be written as follows:

$$\begin{cases} u_1v_1 + u_2v_2 + u_3v_3 = 0 \\ u_1^2 + u_2^2 + u_3^2 = v_1^2 + v_2^2 + v_3^2 = 1 \end{cases}. \tag{1}$$

The points of coordinates $[u, v]^T$ in the measuring plane can be expressed in the camera coordinate system as the vector below,

$$u \begin{bmatrix} u_1 \\ u_2 \\ u_3 \end{bmatrix} + v \begin{bmatrix} v_1 \\ v_2 \\ v_3 \end{bmatrix} + \begin{bmatrix} x_0 \\ y_0 \\ z_0 \end{bmatrix}. \tag{2}$$

Assuming that the coordinate of the point in the original imaging plane is $[x, y, 0]^T$, $\exists k \in R$ the following expression can be derived,

$$u \begin{bmatrix} u_1 \\ u_2 \\ u_3 \end{bmatrix} + v \begin{bmatrix} v_1 \\ v_2 \\ v_3 \end{bmatrix} + \begin{bmatrix} x_0 \\ y_0 \\ z_0 \end{bmatrix} - \begin{bmatrix} 0 \\ 0 \\ f \end{bmatrix} = k \left( \begin{bmatrix} 0 \\ 0 \\ f \end{bmatrix} - \begin{bmatrix} x \\ y \\ 0 \end{bmatrix} \right). \tag{3}$$

Comparing the preceding formula, it yields

$$-k \begin{bmatrix} x \\ y \end{bmatrix} = u \begin{bmatrix} u_1 \\ u_2 \end{bmatrix} + v \begin{bmatrix} v_1 \\ v_2 \end{bmatrix} + \begin{bmatrix} x_0 \\ y_0 \end{bmatrix} = \begin{bmatrix} u_1 & v_1 & x_0 \\ u_2 & v_2 & y_0 \end{bmatrix} \begin{bmatrix} u \\ v \\ 1 \end{bmatrix} \tag{4}$$

$$kf = uu_3 + vv_3 + z_0 - f = \begin{bmatrix} u_3 & v_3 & z_0 - f \end{bmatrix} \begin{bmatrix} u \\ v \\ 1 \end{bmatrix}. \tag{5}$$

Equation (5) can be rewritten as the following,

$$-k = \begin{bmatrix} -\frac{u_3}{f} & -\frac{v_3}{f} & -\frac{z_0 - f}{f} \end{bmatrix} \begin{bmatrix} u \\ v \\ 1 \end{bmatrix}. \tag{6}$$

Combining Equations (6) and (4), it leads to

$$-k \begin{bmatrix} x \\ y \\ 1 \end{bmatrix} = \begin{bmatrix} u_1 & v_1 & x_0 \\ u_2 & v_2 & y_0 \\ -\frac{u_3}{f} & -\frac{v_3}{f} & -\frac{z_0 - f}{f} \end{bmatrix} \begin{bmatrix} u \\ v \\ 1 \end{bmatrix}. \tag{7}$$

For convenience, a parameter matrix M is introduced, as shown below,

$$M = \begin{bmatrix} u_1 & v_1 & x_0 \\ u_2 & v_2 & y_0 \\ -\frac{u_3}{f} & -\frac{v_3}{f} & -\frac{z_0 - f}{f} \end{bmatrix}. \tag{8}$$

If the focus $[0, 0, f]^T$ is not on the measuring plane, the matrix M is a nonsingular matrix. Under normal working conditions, the focus will not be on the measuring plane, so the matrix M can usually be treated as a nonsingular matrix. The focal length and spatial position of the camera will change when the camera moves to a new position to capture the target structure. It can also be considered that the camera imaging plane is fixed, the focal length and the actual spatial position of the structure are changed. Make the coordinates of the camera focus change to $[0, 0, f']^T$, and the original coordinates of the measuring plane change to $\left[ x_0', y_0', z_0' \right]^T$. The unit vectors of the $x, y$ axes in the measuring plane become $\left[ u_1', u_2', u_3' \right]^T, \left[ v_1', v_2', v_3' \right]^T$. Similarly, there is $\exists k \in R$, making the coordinate point $[u, v]^T$ on the measuring plane, and its corresponding imaging point $[x', y', 0]^T$ should satisfy:

$$-k' \begin{bmatrix} x' \\ y' \\ 1 \end{bmatrix} = \begin{bmatrix} u_1' & v_1' & x_0' \\ u_2' & v_2' & y_0' \\ -\frac{u_3'}{f'} & -\frac{v_3'}{f'} & -\frac{z_0' - f'}{f'} \end{bmatrix} \begin{bmatrix} u \\ v \\ 1 \end{bmatrix}. \tag{9}$$

The parameter matrix M′ is denoted as:

$$M' = \begin{bmatrix} u_1' & v_1' & x_0' \\ u_2' & v_2' & y_0' \\ -\frac{u_3'}{f'} & -\frac{v_3'}{f'} & -\frac{z_0' - f'}{f'} \end{bmatrix}. \tag{10}$$

Comparison of Equations (7) and (9) leads to

$$-k' \begin{bmatrix} x' \\ y' \\ 1 \end{bmatrix} = M' \begin{bmatrix} u \\ v \\ 1 \end{bmatrix} = -kM'M^{-1} \begin{bmatrix} x \\ y \\ 1 \end{bmatrix} \tag{11}$$

assuming:

$$M' \cdot M^{-1} = \begin{bmatrix} m_{11} & m_{12} & m_{13} \\ m_{21} & m_{22} & m_{23} \\ m_{31} & m_{32} & m_{33} \end{bmatrix}. \tag{12}$$

Accordingly, the following expansions are introduced:

$$\begin{cases} k'x' = k(m_{11}x + m_{12}x + m_{13}) \\ k'x' = k(m_{21}x + m_{22}x + m_{23}) \\ k' = k(m_{31}x + m_{32}y + m_{33}) \end{cases}. \tag{13}$$

Therefore, there is:

$$\begin{cases} x' = \frac{m_{11}x + m_{12}x + m_{13}}{m_{31}x + m_{32}y + m_{33}} \\ y' = \frac{m_{21}x + m_{22}x + m_{23}}{m_{31}x + m_{32}y + m_{33}} \end{cases}. \tag{14}$$

The coordinates $(x, y)$ are the imaging point of the original image, which is transformed into a new imaging point $(x', y')$ after the perspective transformation. Based on the above analysis, the proposed process can convert the original oblique structural image into orthophoto-projection image, which provides technological foundation for monitoring the overall deformation of the structure.

### 2.2. Edge Detection

The edge of structural image is an important carrier of overall deformation information. Edge detection is a method to analyze the main features of images [21]; it can greatly reduce the amount of data, eliminate information not related to deformation monitoring, and retain the basic attributes of structure. The basic task of edge detection is to recognize the step change of the gray value of the structure edge in the image, which can be further used to obtain the feature edge of the structure. According to [22], the step edge is related to the peak value of the first-order derivative of the gray level of the image, and the degree of change of the gray value can be expressed by gradient. The gradient of image function is a vector with direction and size, as shown below:

$$G(x, y) = \begin{bmatrix} G_x \\ G_y \end{bmatrix} = \begin{bmatrix} \frac{\partial f}{\partial x} \\ \frac{\partial f}{\partial y} \end{bmatrix}. \tag{15}$$

It can be seen that the direction of the vector $G(x, y)$ is the change rate of the gray value of function $f(x, y)$.

The amplitude of the gradient can be expressed as:

$$\left| G(x, y) \right| = \sqrt{G_x^2 + G_y^2}. \tag{16}$$

In this paper, the absolute value is used to approximate the gradient amplitude:

$$\left| G(x, y) \right| \approx \max \left( |G_x|, |G_y| \right). \tag{17}$$

The direction of the gradient can be derived as:

$$\alpha(x, y) = arctan(G_y/G_x) \tag{18}$$

where $\alpha$ is the angle between the direction vector and the x-axis.

From the above formulas, it is suggested that the degree of the change in gray levels can be detected by the discrete approximation function of gradient. At the edge of the structure, the gray value will change [23], resulting in the maximum value of the gradient function. On this end, the edge can be extracted through the above features.

From the above algorithm, we can see that the essence of the image edge is the point of discontinuous gray level, or where the gray level changes dramatically. The drastic change of gray level of edge means that near the edge point, the signal has high frequency components in the spatial domain. Therefore, the edge detection method is essentially to detect the high-frequency component of the signal, but it is difficult to distinguish the high-frequency component of the gray signal from the environmental noise of the actual structure photogrammetry, which makes it difficult to accurately extract the edge information of the structure. Taking one-dimensional signal of the structure image as an example, as shown in Figure 2, if point A is regarded as the edge point of the signal and there is a jump in the signal, then whether there is an edge at point B and point C needs to be treated with caution. In fact, point B and point C are probably the combination of the signal and some noise.

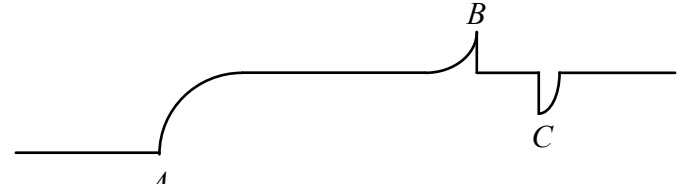

**Figure 2.** Illustration of the edge point and noise point.

Edges with continuous gradients, such as point A in Figure 2, are very rare in actual structure images. Most of the structural edge points will be accompanied by environmental noise, forming a large number of complex edge points such as points B and C. Therefore, it is necessary to study the false edges caused by noise in order to ensure the accuracy of structural deformation monitoring.

## 3. Static Test of the Beam

A static test has been carried out on a steel truss–concrete composite beam, to validate the proposed method, as shown in Figure 3.

The specimen is simply supported by two hinge bearings at the both ends, as shown in Figure 3a. Two hydraulic jacks have been used to apply the two-points bending load on the specimen. Three dial meters have been placed at the quarter-span and the midspan of the specimen, to measure the structural deflection.

The specimen has been loaded with a step-by-step prototype from 0 to 600 kN, with an increment of 100 kN. The loading protype is shown in Figure 4. It is worth stating that the measurement at each step has been made two minutes after the target load is reached, to allow the well-deformation of the specimen.

During the test, the digital image of the specimen is also collected using Canon EOS 5DS R low-cost digital camera; the camera and lens parameters are shown in Table 1. The spatial position of the camera in this experiment is set in the non-orthogonal projection position to simulate the normal condition in engineering practice, as aforementioned. During the whole process, the space position and azimuth of the camera should be maintained to ensure the consistency of projection centers of structure images in the whole process. In order to prevent the camera from being disturbed, shooting remote controller is used to control camera parameters and shutter shooting.

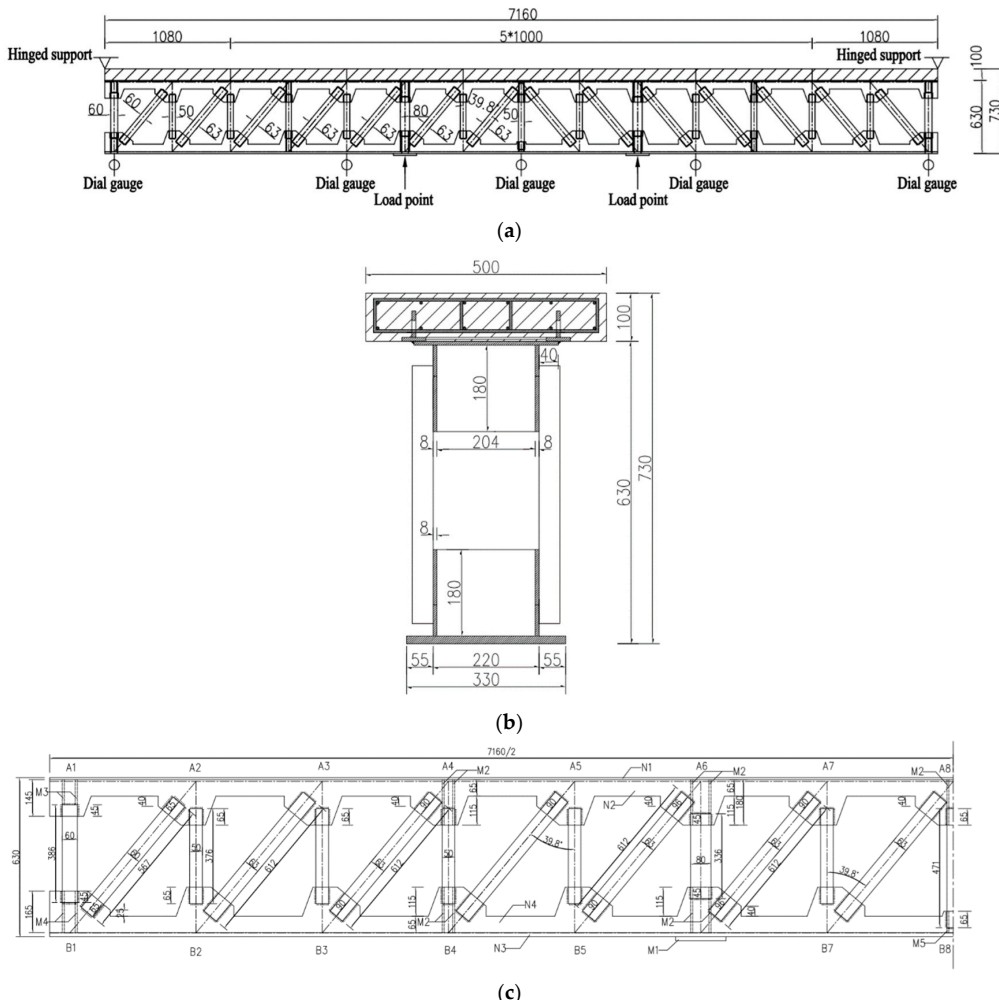

**Figure 3.** The tested steel–concrete composite beam (unit: mm); (**a**) elevational view; (**b**) sectional view; (**c**) detail size.

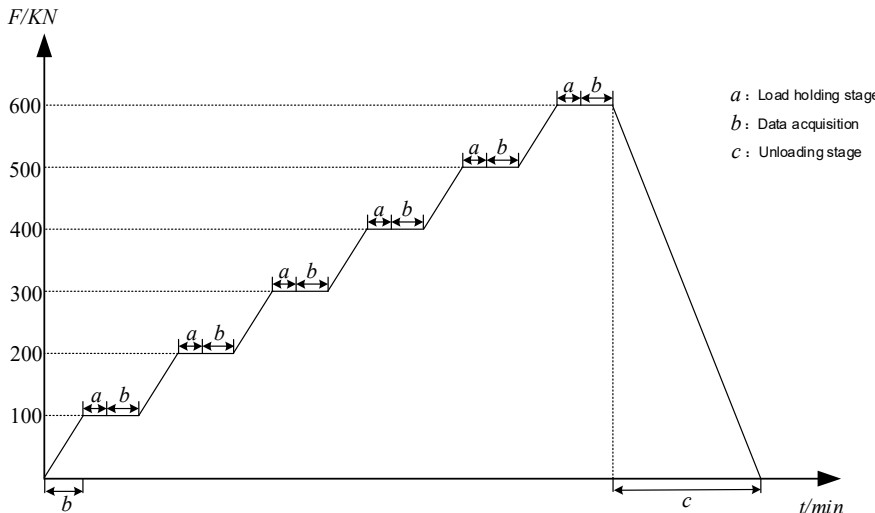

**Figure 4.** Test loading protype.

**Table 1.** Parametric table of camera and lens.

| Number of Pixels | Size of Sensor | Data Interface | Aspect Ratio | Photo-Sensors |
|---|---|---|---|---|
| 50.6 million | $36 \times 24$ mm | USB 3.0 | 3:2 | CMOS |
| **Image Amplitude** | **Pixel Size** | **Lens Type** | **Focal Length** | **Lens Relative Aperture** |
| $8688 \times 5792$ | 4.14 μm | EF 24–70 mm f/2.8 L | 50 mm | F2.8–F22 |

As a common practice, the system error exists in the measurement due to the physic limitation of the applied hardware. Specifically, the accuracy of the photogrammetry-based method has a stronger dependence on the capacity of hardware when compared with the traditional methods. Therefore, the calibration of photogrammetry equipment is an essential part of the measurement. Generally, the largest part of error in photogrammetric hardware originates from lens distortion [24]. On this end, the checkerboard lattice calibration method [25,26] has been applied to calibrate the lens of photogrammetric camera. The calibration has been conducted with a total of 25 checkerboard lattice images, and the lens distortion parameters are obtained. Based on that, the photogrammetric images obtained in this paper have been corrected. The calibration process is shown in Figure 5.

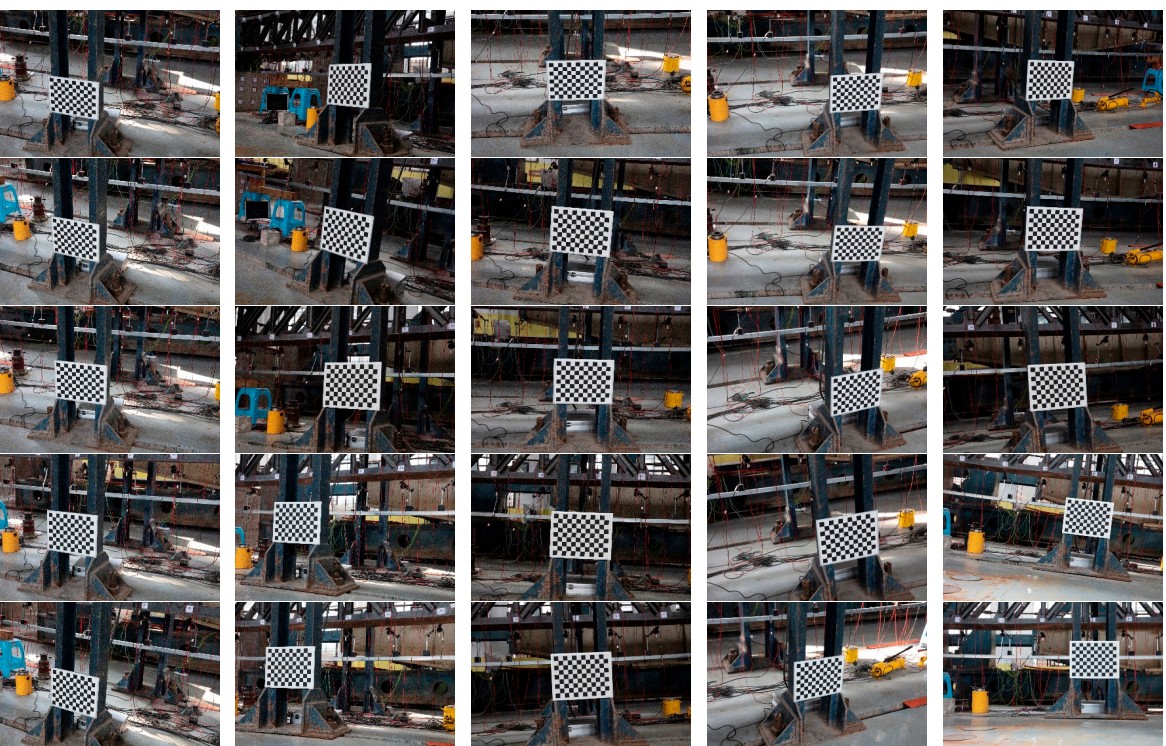

**Figure 5.** Camera calibration intersection photography.

The results of camera calibration parameters are shown in Table 2.

**Table 2.** Parametric table of camera and lens.

| Outline Size | Phase Principal Coordinates X0 | −0.0769 mm |
| | Phase Principal Coordinates Y0 | 0.0045 mm |
| | Camera Main Distance f | 50.9339 mm |
| Radial Distortion Coefficient | K1 | $1.9644 \times 10^{-8}$ |
| | K2 | $5.6287 \times 10^{-6}$ |
| Eccentric Distortion Coefficient | P1 | $1.4683 \times 10^{-5}$ |
| | P2 | $-3.8601 \times 10^{-6}$ |
| Pixel Size | 0.004096 mm | |
| Image Size | $5792 \times 8668$ | |

In the calibration, the distortion correction formula [27] has been used to correct the structural image element by element. As a result, the ideal image without lens distortion has been obtained, which can be further used for the extraction of structural deformation in the next. The distortion correction effect of the image is shown in Figure 6.

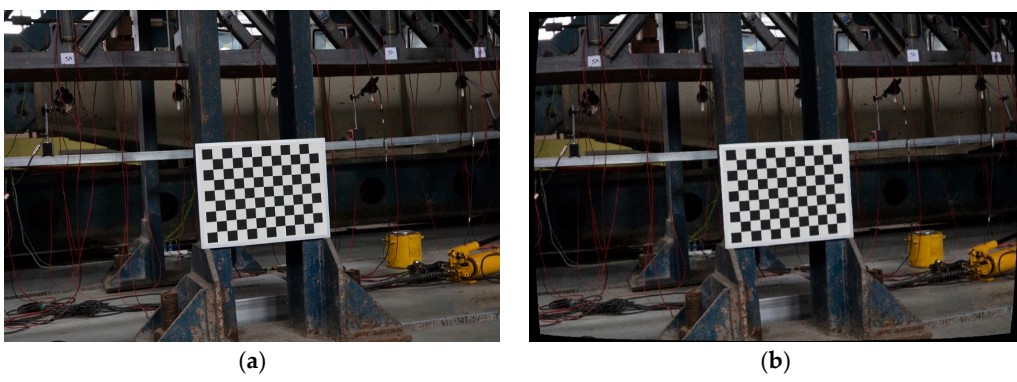

(**a**) (**b**)

**Figure 6.** Camera calibration distortion adjustment; (**a**) original image; (**b**) image after distortion correction.

In reality, the landform around the structure makes it difficult to obtain the orthogonal projection image, so that it can only be tilt photographed on both sides. According to the practical application requirements, the actual situation is simulated in the test, i.e., the camera takes pictures of the test beam at a fixed tilt angle. As shown in Figure 7a, the distance between the camera and the ground is about 3.5 m, and the distance between the camera and the test beam is about 3.0 m. Obviously, there is a large horizontal angle between the optical axis of the camera and the normal direction of the vertical plane of the test beam, and there is also an elevation angle in the vertical direction. This photogrammetric method simulates the possible inclination angle of the camera in the actual structure survey; unlike orthographic projection, this tilt photography will cause the structure image to be affected by the perspective relationship and present near-large-far-small imaging features, which will affect the extraction of structural deformation information. Figure 7 shows the image acquisition result of the specimen. It is worth noting in the image that the specimen near the right bearing is blocked by the reaction frame, which can reflect the usual monitoring conditions of actual structure. Thus, it is crucial to obtain the deformation data of the sheltered part of the structure, which is also a key part of the present study.

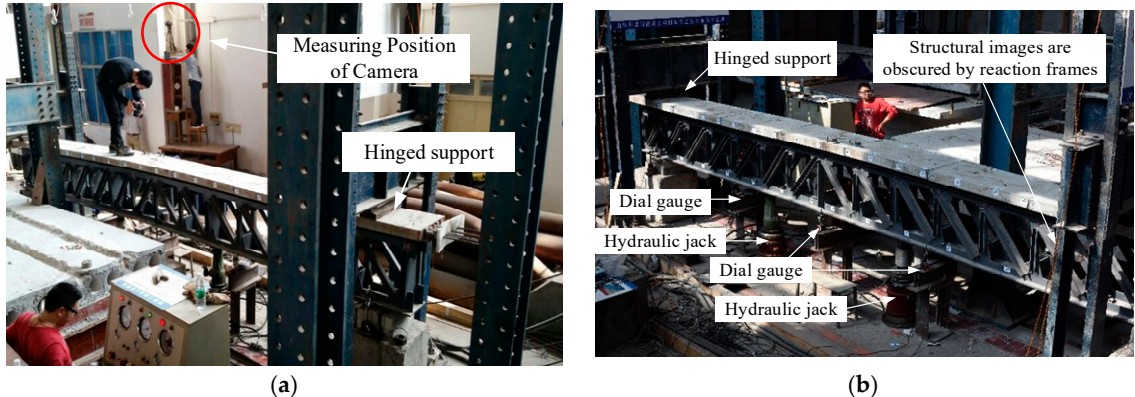

**Figure 7.** Field layout of static load test; (**a**) camera placement position; (**b**) collecting shape data of experimental beam.

## 4. Result and Discussion

### 4.1. Picture Perspective Transform of Test Beam

According to the principle of perspective transformation, the image of the test beam obtained under each load grades is processed, and the result of image processing under one of the load grades is shown in Figure 8. It can be seen that through perspective transformation, the image of the test beam changes from oblique projection to orthogonal projection, while all the details of the specimen have been well preserved. It is worth noting that the remain parts apart from the specimen are distorted by the perspective transformation. However, the distortion does not affect the acquisition of structural deformation data since the test aims at the overall deformation data of the specimen only.

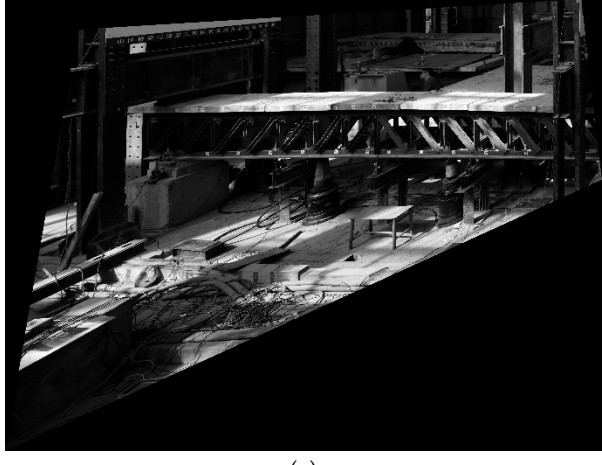

(**a**)

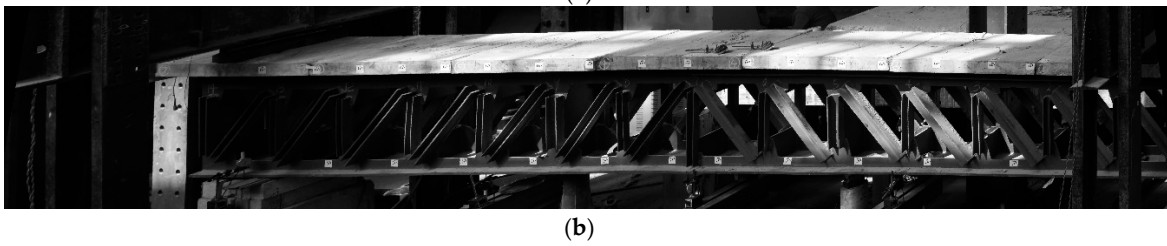

(**b**)

**Figure 8.** Perspective transformation of the tested beam; (**a**) after perspective transform; (**b**) integral drawing of specimen after projection transformation.

### 4.2. Edge Contour Extraction of Structures

Several types of operators can be employed for the edge detection, including the Sobel [28] operator, Prewitt operator [29], Roberts operator [30], and log operator [31], in which different methods are used to solve the gradient extremum. In this paper, all the five operators have been applied to detect the edge of the specimen using the image after perspective transformation, as shown in Figure 9. Compared with the Log operator, the edge detection results of the other four operators are not satisfactory due to the lack of edge information, which in turn has a negative impact on the accuracy. The advantage of the Log operator over the other methods is that the Gauss spatial filter is employed to smooth the original image, which minimizes the influence of noise on edge detection. The Log operator is a second-order edge detection operator [32], as shown in Equations (19) and (20):

$$\nabla^2 f = \frac{\partial^2 f}{\partial x^2} + \frac{\partial^2 f}{\partial y^2} \tag{19}$$

$$\begin{cases} \frac{\partial^2 f}{\partial x^2} = f(i, j+1) - 2f(i,j) + f(i, j-1) \\ \frac{\partial^2 f}{\partial y^2} = f(i+1, j) - 2f(i,j) + f(i-1, j) \end{cases} . \tag{20}$$

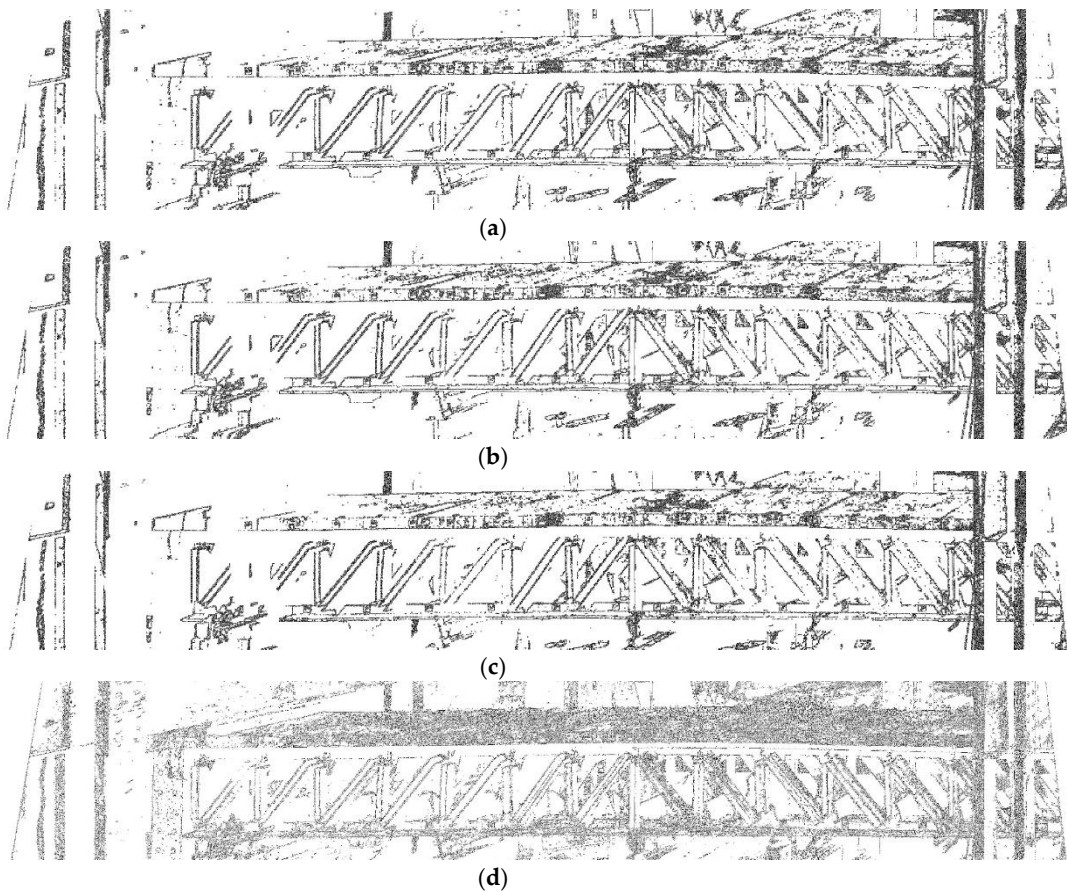

(a)

(b)

(c)

(d)

**Figure 9.** The applied five kinds of edge detection operators; (**a**) Sobel operator edge detection; (**b**) Prewitt operator edge detection; (**c**) Roberts operator edge detection; (**d**) log operator edge detection.

Based on the second-order differential of the image, the extreme points can be generated at the abrupt position of the gray value. According to these extreme points, the edge of the structure can be determined.

As shown in Figure 9, the distribution of the light intensity respecting the specimen is inconsistent, and the gray value of some edges does not change significantly, resulting in discontinuity in the detection of some edges. The discontinuous edges like Figure 9 are very normal in actual structure images. On the other hand, the distribution of edge pixels obtained by various edge algorithms is also different. The distribution density of edge pixels in this paper is shown in Figure 10.

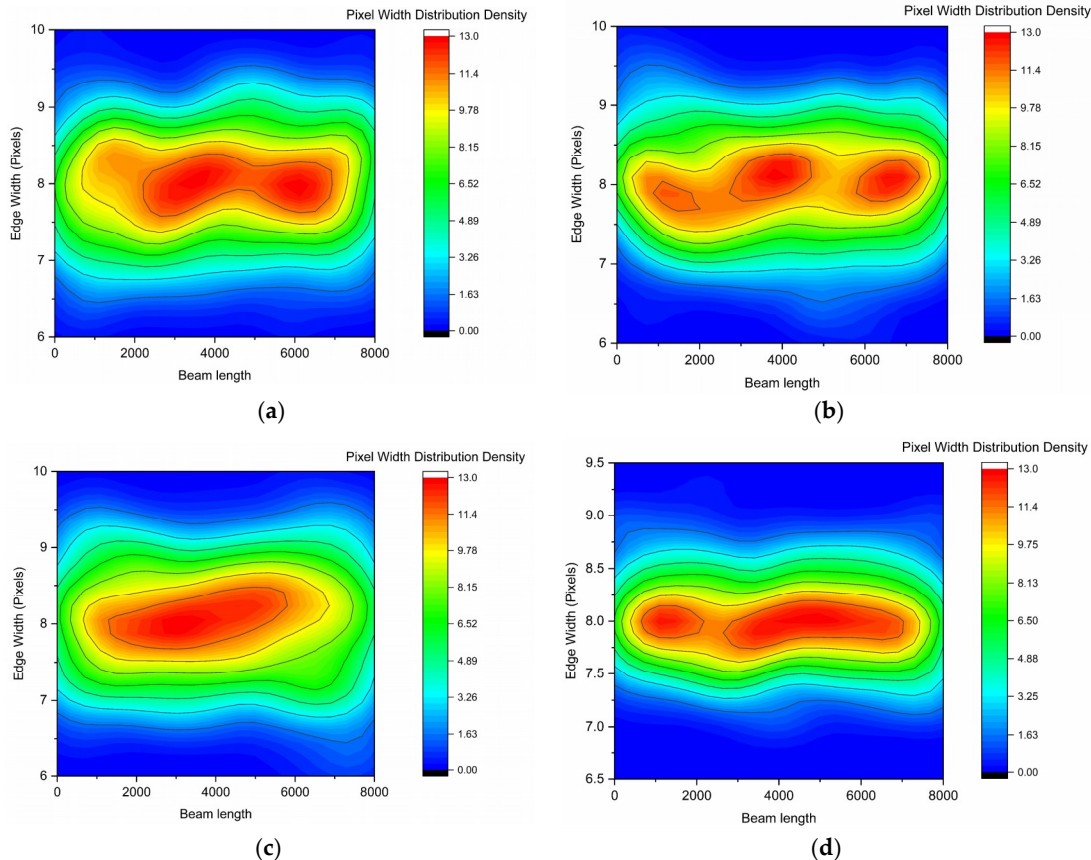

**Figure 10.** Distribution density of edge pixels by various edge detection operators; (**a**) edge width distribution degree of Sobel operator; (**b**) edge width distribution degree of Prewitt operator; (**c**) edge width distribution degree of Robert operator; (**d**) edge width distribution degree of log operator.

As shown in Figure 10, the edge features obtained by the above five operators are entirely different. For instance, the edge distributions in some operators are three pixels wide, while in some operators, e.g., the Log operator, the edges are just concentrated in one pixel. Because of the large scatter in the pixel distribution, it is difficult to determine the exact edge position of the structure. Naturally, an efficient edge detection operator should have the centralized pixel distribution and good pixel continuity. After comparison, it is found that the Log operator can extract the edges of the structure relatively intact and maintain a relatively centralized distribution of edge pixels. Thus, the log operator has been selected in extracting the edge of the structure.

The phenomenon of discontinuous edges and scattered edge distribution of the above-mentioned is similar to the detection issue in real structures, which is induced by the environment of measurement. In dealing with such kind of problem, the data processing and analysis process have been employed, as illustrated in the following. As an important part of the deformation, the edge of the structure is the key content of this paper. For the specimens in this paper, the upper and lower edges of the bridge deck and the lower edges of the specimens can be used as characteristic contours to analyze the overall deformation of the structure. From Figure 2, we can see that the noise caused by the environment will make the edge location confused. Points with continuous gradient change are very rare in the actual

structure image. The presence of image noise can lead to the generation of pseudo edges. On the other hand, the real signal on the edge of the structure may also be smoothed out by the Gauss spatial filter, which will cause the edge of the structure to be discontinuous and the edge information missing as shown in Figure 11. Figure 11a shows that the lower edge contour of the bridge deck is relatively continuous. Therefore, the lower edge of the bridge deck is used as the characteristic contour of the test beam to extract the overall deformation of the structure; the position of edge contour extraction in this paper is shown in Figure 11b.

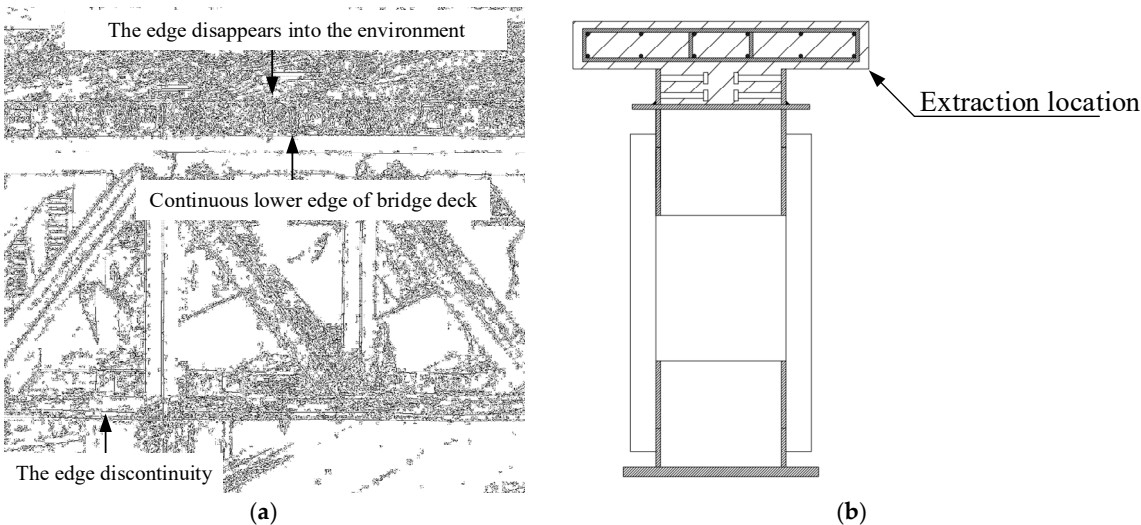

(**a**)　　　　　　　　　　　　　　　　　　　　　(**b**)

**Figure 11.** Feature contour extraction; (**a**) edge of the test beam affected by environment; (**b**) sketch map of edge extraction position.

Because the space position of the camera is fixed and the same perspective transformation method is used in each load grade, the edge contour before and after deformation can be directly extracted and compared without looking for fixed points in each load grade. The pixel coordinates of the lower edge of the bridge deck are extracted, and the original edge contour of the structure is obtained under each load grade, as shown in Figure 12.

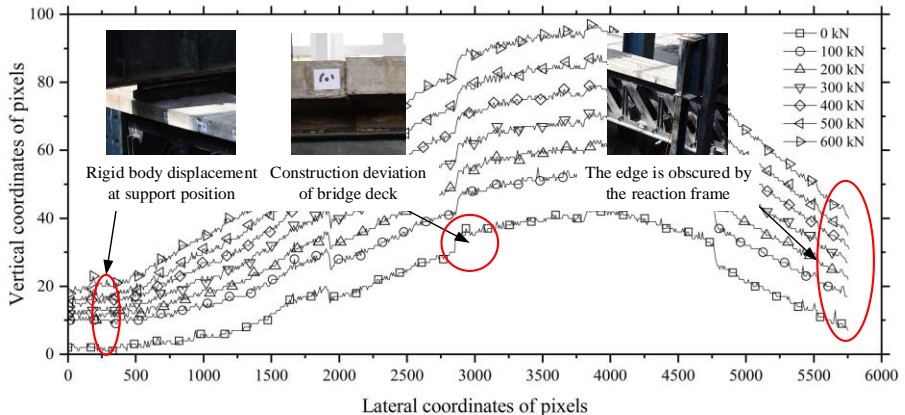

**Figure 12.** Edge line of lower edge of bridge deck.

The result shows that the deflection of the specimen increases with the load grade. Since the edge line of the specimen is obscured near the right bearing by the reaction frame, the edge information is partially missed. By contrast, the information of the left edge is well established. Considering the parallel change of the contour line near the bearing on the left, it can be inferred that the rigid displacement of the specimen exists under each load condition. When loaded from 0 to 100 kN,

the rigid displacement reaches its largest. This large displacement is due to the fact that the bearings are not in close contact with the reaction frame before loading. After the load of 100 kN, the bearings will be tightly contacted with the reaction frame. However, due to the deformation of the reaction frame since it is not ideally stiff, a small amount of rigid body displacement still exists in all the load conditions. Besides, it can be found that the edge contour of the specimen is piecewise continuous under every single load condition, with the step change occurred at the four connections between the segments. During the fabrication, the specimen is first divided into five segments and each of the segments is manufactured independently. Then, the separated segments are assembled at the four points together, resulting in the inevitable assembly error. As a result, the shape of the specimen will change abruptly at those assembly points, which in turn lead to the step change as reflected in the edge contour.

### 4.3. Deformation Curve Obtained by Overlapping Difference of Contour Line

By calibrating each pixel in the image, the size of each pixel can be obtained. The size of each pixel is the theoretical limit of the accuracy in the proposed photogrammetric method. In the steel truss bridge tested, the vertical member is orthogonal to the pixels, as shown in Figure 13. Thus, the calibration of the vertical member is relatively simple. As shown in Table 3, by calibrating 13 visible vertical members, the measurement accuracy of this experiment is 1.12 mm. According to the calibration value, the contour of the pixel in Figure 12 can be transformed into the actual deformation value.

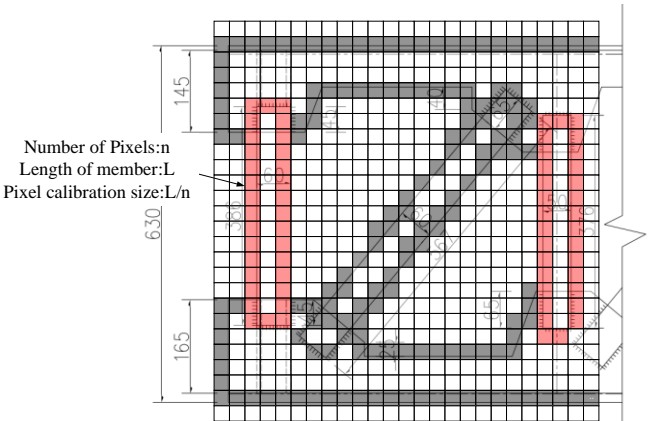

**Figure 13.** Pixel calibration schematic.

**Table 3.** Pixel size calibration table of vertical members.

| Number of Vertical Members | Number of Pixels | Real Length of Members/mm | Calibration Value | Average Value |
|---|---|---|---|---|
| 1 | 344 | 387 | 1.12 | |
| 2 | 342 | 379 | 1.10 | |
| 3 | 340 | 380 | 1.11 | |
| 4 | 345 | 377 | 1.09 | |
| 5 | 339 | 376 | 1.10 | |
| 6 | 292 | 332 | 1.12 | |
| 7 | 346 | 375 | 1.08 | 1.12 mm/px |
| 8 | 421 | 473 | 1.12 | |
| 9 | 330 | 376 | 1.14 | |
| 10 | 295 | 331 | 1.22 | |
| 11 | 335 | 377 | 1.12 | |
| 12 | 351 | 376 | 1.07 | |
| 13 | 342 | 381 | 1.11 | |

The edge line shown in Figure 12 is notably discontinuous at the connecting points due to assembly error as analyzed before. However, the discontinuity is not caused by the structural deformation and will not change with the applied load. Therefore, the influence of the discontinuity can be eliminated by the overlapping difference method. Since the bearings contact the reaction frame closely after 100 kN, the edge line of the initial working condition can be overlap differenced by the edge line of the other load grades. Thus, the load-displacement curve of 100–600 kN can be derived, as shown in Figure 14.

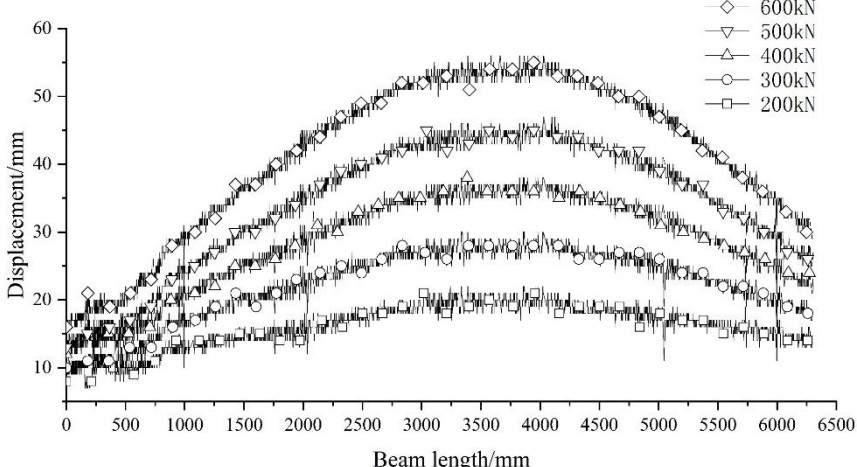

**Figure 14.** Load-displacement curves under various load grades.

It can be found that the load-displacement curve presents a zigzag feature. The reason for this problem is that the edge of the structure in the image is composed of one or more pixels, as shown in the red pixel of Figure 15. These pixels are arranged side by side to form a bandwidth. The final edge position (shown in deep red in Figure 15) is determined by the gradient change rate of the gray value of all pixels in the bandwidth. Under the influence of illumination conditions, the final edge and the actual edge will have errors, as shown in the dotted line of Figure 15. As a result, the zigzag phenomenon occurs in the load-displacement curve shown in Figure 14. Based on the above analysis, it can be indicated that the deformation extracted from the image will be distributed around the actual deformation of the structure. On this end, the load-deformation curve is polynomial fitted to approximate the actual deformation value of the structure, as shown in Figure 16. The measured comparison of three dial meters is shown in Figure 17.

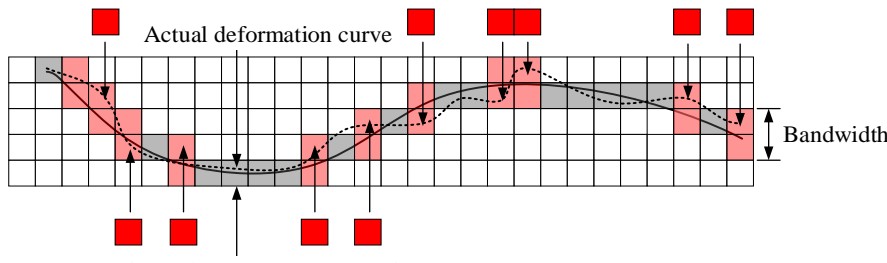

**Figure 15.** Difference between pixel edge and actual edge.

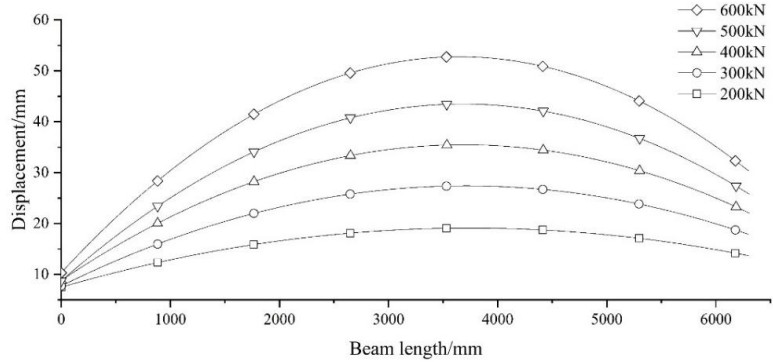

**Figure 16.** The load-displacement curve after fitting.

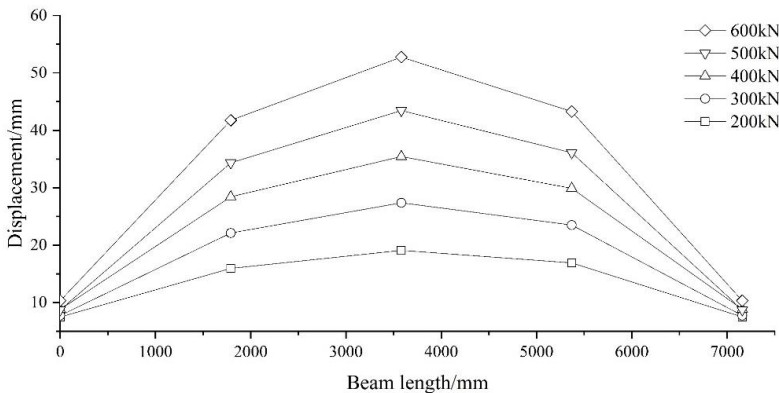

**Figure 17.** The load-displacement curve measured by dial meters.

### 4.4. Error Analysis of Photogrammetry

The overall deformation of the structure edge extracted by photogrammetry is compared with the dial gauge as shown in Figure 18. The numerical comparison results are shown in Table 4. From Figure 18 and Table 4, it can be seen that the structural deformation data obtained by photogrammetry are consistent with the data measured by the dial meters, and the maximum error is less than 5%. Compared with the traditional methods, the photogrammetry method has a wider observation range. Thus, the proposed method can obtain the displacement of any section without extensive efforts, which in turn can better reflect the overall displacement and deformation of the structure.

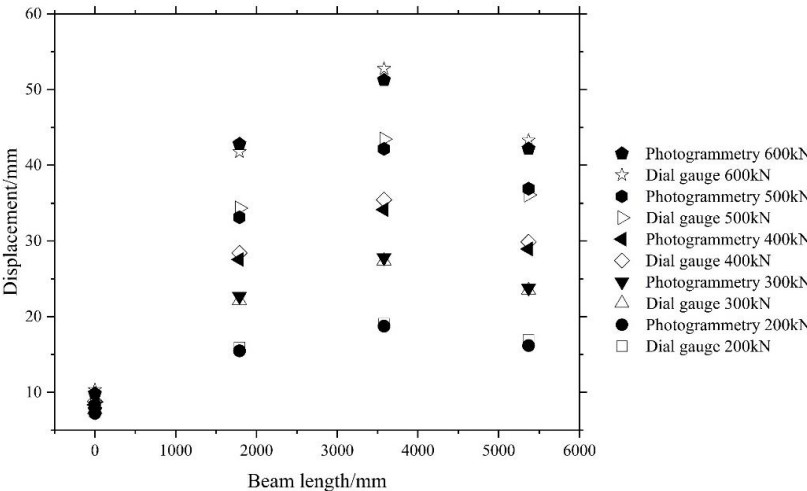

**Figure 18.** Error comparison chart.

**Table 4.** Comparison of displacement measurement error.

| Loading Condition/kN | Dialgauge Position | Dialgauge Measured Value R1/mm | Photogrammetric Value R2/mm | \|R2 − R1\| = S/mm | Error S/R1/% | RMSE |
|---|---|---|---|---|---|---|
| 200 | Left support | 7.50 | 7.25 | 0.25 | 3.33 | |
| | L/4 | 15.95 | 15.50 | 0.45 | 2.82 | |
| | 2L/4 | 19.09 | 18.75 | 0.34 | 1.78 | |
| | 3L/4 | 16.90 | 16.19 | 0.71 | 4.20 | |
| 300 | Left support | 7.74 | 7.45 | 0.29 | 3.75 | |
| | L/4 | 22.13 | 22.69 | 0.56 | 2.53 | |
| | 2L/4 | 27.37 | 27.80 | 0.43 | 1.57 | |
| | 3L/4 | 23.49 | 23.78 | 0.29 | 1.23 | |
| 400 | Left support | 8.82 | 8.47 | 0.35 | 3.97 | 0.82 |
| | L/4 | 28.42 | 27.56 | 0.86 | 3.03 | |
| | 2L/4 | 35.45 | 34.15 | 1.30 | 3.67 | |
| | 3L/4 | 29.89 | 28.94 | 0.95 | 3.18 | |
| 500 | Left support | 8.72 | 8.36 | 0.36 | 4.13 | |
| | L/4 | 34.34 | 33.14 | 1.20 | 3.49 | |
| | 2L/4 | 43.46 | 42.18 | 1.28 | 2.95 | |
| | 3L/4 | 36.08 | 36.91 | 0.83 | 2.30 | |
| 600 | Left support | 10.33 | 9.84 | 0.49 | 4.74 | |
| | L/4 | 41.76 | 42.81 | 1.05 | 2.51 | |
| | 2L/4 | 52.75 | 51.26 | 1.49 | 2.82 | |
| | 3L/4 | 43.28 | 42.19 | 1.09 | 2.52 | |

## 5. Conclusions

Based on digital image processing technology, this paper presents a novel method for structural overall deformation monitoring. Using the proposed method, a series of deformation measurement experiments are carried out on a steel–concrete composite beam, and the main conclusions are as the following:

(1) Due to the limitation on camera sites, orthogonal projection images are usually difficult to be accessed for large engineering structures such as bridges. In dealing with the issue, the perspective transformation method is applied to acquire the orthogonal projection of structures from the originally inclined images. The experimental results show that the orthogonal projection image obtained by the proposed method can correctly reflect the overall deformation of the structure.

(2) In order to characterize the key feature of structural deformation, the edge detection operator is utilized to obtain the edge contour of the structure from the processed orthogonal images. Using the operator, the overall deflection curve of the structure can be obtained by locating and calibrating the edge pixels.

(3) The edge line of the structure acquired from the position of the pixel shows a notable zigzag effect. Further investigations have been carried out, and the result suggests that the illumination environment can be mainly attributed to the zigzag effect. Since the image edge of the structure has a certain bandwidth, the final position of edge pixels in the bandwidth range will be affected by the illumination environment, which eventually results in the fluctuation around the actual deflection curve. On this end, the fitting method is used to minimize the fluctuation and obtain the linear approximation of the actual deflection curve. After comparison with the data measured by the dial meters, it shows the error of the proposed method is less than 5%.

(4) Since the proposed method is based on digital images, the accuracy is dependent on the quality of available images even if some advanced image processing methods are utilized. For instance, a major limitation of the method is that the overall deformation cannot be directly obtained when

some parts of the measuring structures are obscured. Under such a situation, the postprocessing method, such as the fitting, can be applied to obtain the approximation data of the blocked parts.

**Author Contributions:** X.C. conceived this study, designed the computational algorithms, wrote the program code, and wrote the manuscript. Z.Z. proposed some valuable suggestions and guided the experiments. G.D., X.D. acquired the test images and performed the experiments. X.J. carried out the measurements and analyzed the experimental data.

**Funding:** This research was funded by the National Natural Science Foundation Projects (Grant No.: 51778094, 51708068).

**Acknowledgments:** Special thanks to J.L. Heng at the Southwest Jiaotong University.

**Conflicts of Interest:** The authors declare no conflict of interest.

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
