# Peer review of "An Overall Deformation Monitoring Method of Structure Based on Tracking Deformation Contour"

_applsci, doi:10.3390/app9214532_

Round 1

Author Response

The authors highly appreciate all the kind comments from the reviewer, which help to improve our manuscript significantly.

Reviewer 2 Report

The paper deals with the deformation monitoring of structures using photogrammetry. For this reviewer structural deformation monitoring is an important part of structural health monitoring, therefore this paper is of high interest. However, some major clarifications/improvements are required.

Photogrammetry is a technology well known and used for structural deformation monitoring. Therefore, the authors are strongly encouraged to highlight the novelty aspects of their research. Moreover, it would be interesting to know which are the differences/novelty and advantages of their approach if compared with approaches already known as: 1) Detchev, I., Habib, A., & El-Badry, M. (2011, May). Case study of beam deformation monitoring using conventional close range photogrammetry. In ASPRS 2011 Annual Conference. ASPRS. Milwaukee, Wisconsin, USA; 2) Artese, S., Achilli, V., & Zinno, R. (2018). Monitoring of bridges by a laser pointer: Dynamic measurement of support rotations and elastic line displacements: Methodology and first test. Sensors18(2), 338.

Finally, the use of the proposed approach can introduce errors in the measurements. This aspect is not well described in the manuscript, as well as no magnitude of the errors are provided. The authors are encourage to provided more insights on this aspect.

Author Response

(The authors gave the same response as above.)

Reviewer 3 Report

see attached document

Author Response

The authors highly appreciate all the kind comments from the reviewer, which help to improve our manuscript a lot.

Round 2

Reviewer 2 Report

The paper can be accepted in the present form. 

Author Response

A throughout check has been performed on the language in the revised manuscript. The authors highly appreciate all the kind comments from the reviewer, which help to improve our manuscript a lot.

Reviewer 3 Report

The quality and soundness of the manuscript is improved significantly, well done. The authors have very carefully and professionally addressed all my comments. The answer document itself is as long as the paper. I am also happy that the authors shared a little on their project (new publication) on 3D reconstruction of the structure. I will look up for it.

I have only a few minor comments. It is my opinion that the paper is ready for publications

Overall, the manuscript is well written, however, I still recommend it to be proofread by a native speaker. L259 a reference is missing. The presentation of Figure 10 (you have named it Figure 3 see L334) could be improved. I recommend putting two images in a row and use a single pixel width distribution density bar for all surface plots. This might be a little trick to accomplish, but it would look better. Check figure numbers.

Author Response

The authors are grateful for reviewer affirmation of the manuscript. Once again, we have seriously revised the manuscript. The numbers of graphs and tables were checked, added references, pixel width distribution density bar are unified, and the arrangement of graphs are adjusted. A careful check has been performed throughout the manuscript to correct the typos again.

The authors highly appreciate all the kind comments from the reviewer, which help to improve our manuscript significantly.
